# Multiple protein-DNA interfaces unravelled by evolutionary information, physico-chemical and geometrical properties

**Flavia Corsi[1,2], Richard Lavery[3], Elodie Laine[1]\*, Alessandra Carbone[1,4]\***

**1** Sorbonne Université, CNRS, IBPS, UMR 7238, Laboratoire de Biologie Computationnelle et Quantitative (LCQB), 75005 Paris, France, **2** Sorbonne Université, CNRS, Institut des Sciences, du Calcul et des Données (ISCD), 75005 Paris, France, **3** Lyon University, CNRS, IBCP, UMR 5086, Molecular Microbiology and Structural Biochemistry, 69367 Lyon, France, **4** Institut Universitaire de France, 75005 Paris, France

\* elodie.laine@upmc.fr (EL); alessandra.carbone@lip6.fr (AC)

**Data Availability Statement:** The data underlying the results presented in the study are available from http://www.lcqb.upmc.fr/JET2DNA and http://www.lcqb.upmc.fr/PDNAbenchmarks.

## Abstract

Interactions between proteins and nucleic acids are at the heart of many essential biological processes. Despite increasing structural information about how these interactions may take place, our understanding of the usage made of protein surfaces by nucleic acids is still very limited. This is in part due to the inherent complexity associated to protein surface deformability and evolution. In this work, we present a method that contributes to decipher such complexity by predicting protein-DNA interfaces and characterizing their properties. It relies on three biologically and physically meaningful descriptors, namely evolutionary conservation, physico-chemical properties and surface geometry. We carefully assessed its performance on several hundreds of protein structures and compared it to several machine-learning state-of-the-art methods. Our approach achieves a higher sensitivity compared to the other methods, with a similar precision. Importantly, we show that it is able to unravel 'hidden' binding sites by applying it to unbound protein structures and to proteins binding to DNA via multiple sites and in different conformations. It is also applicable to the detection of RNA-binding sites, without significant loss of performance. This confirms that DNA and RNA-binding sites share similar properties. Our method is implemented as a fully automated tool, $JET^2_{DNA}$, freely accessible at: http://www.lcqb.upmc.fr/JET2DNA. We also provide a new dataset of 187 protein-DNA complex structures, along with a subset of 82 associated unbound structures. The set represents the largest body of high-resolution crystallographic structures of protein-DNA complexes, use biological protein assemblies as DNA-binding units, and covers all major types of protein-DNA interactions. It is available at: http://www.lcqb.upmc.fr/PDNAbenchmarks.

## Author summary

Protein-DNA interactions are essential to living organisms and their impairment is associated to many diseases. For these reasons, they have become increasingly important therapeutic targets. Experimental structure determination has revealed different binding

**Funding:** This work was funded by LabEx CALSIMLAB (public grant ANR-11-LABX-0037-01 constituting a part of the "Investissements d'Avenir" program - reference: ANR-11-IDEX-0004-02) (FC) and the Institut Universitaire de France (AC). The funders had no role in study design, data collection and analysis, decision to publish, or preparation of the manuscript.

**Competing interests:** The authors have declared that no competing interests exist.

motifs and modes, associated to different functions. Yet, the available structural data gives us only a glimpse of the multiplicity and complexity of protein surface usage by DNA. In this work, we use a three-layer model to describe and predict DNA-binding sites at protein surfaces. Given a protein, we consider the way its residues are conserved through evolution, their physico-chemical properties and geometrical shapes to decrypt its surface. We are able to detect a large portion of interacting residues with good precision, even when they are 'hidden' by conformational changes. We highlight cases where one protein binds DNA via distinct regions to perform different functions. We are able to uncover the alternative binding sites and relate their properties with their specific roles. Our work can help guiding mutagenesis experiments and the development of new drugs specifically targeting one site while limiting possible side effects.

## Introduction

Interactions between proteins and DNA play a fundamental role in essential biological processes [1] and their impairement is associated with human diseases [2, 3]. Thus, they represent increasingly important therapeutic targets. The experimental determination of protein-DNA complexes is a costly and time consuming process (<5000 protein-DNA complex structures available in the Protein Data Bank [4], release 2019). This has motivated the development of a large number of computational methods, most of them based on machine learning, to predict DNA-binding residues [5–24]. Both sequence-based properties (conservation, amino acid type, predicted secondary structure, solvent accessibility and disorder) and structural properties (electrostatic potential, dipole moment, surface shape and curvature, structural motifs, secondary structure, amino acid microenvironment, solvent accessibility and hydrogen-bonding potential) have been considered. Among those, amino acid composition is one of the most powerful feature. Indeed, positively charged and polar amino acids are largely over-represented in DNA-binding sites, in order to counterbalance the excess of negative charge coming from the DNA phosphate groups [25]. Methods using a large body of features can achieve very high accuracy on known binding sites but generally lack interpretability.

Our understanding of protein-DNA interactions is hampered by the fact that protein surfaces are complex dynamical objects which may interact with several DNA molecules at the same time or at different moments but via the same region [26–28], accommodate indifferently DNA and RNA [29] and undergo substantial conformational changes upon binding [30]. A protein may harbour multiple DNA-binding sites on its surface, each of which may have a different role with a different level of importance in the accomplishment of the protein's function [28, 31] and thus display particular properties. The relatively small amount of available structural data gives us only a glimpse of the many ways nucleic acid molecules use protein surfaces. This calls for the development of tools able to comprehensively identify and characterize protein-DNA interfaces and help decipher protein surface complexity.

Here, we present JET$^2_{DNA}$, a new method for predicting DNA-binding sites at protein surfaces based on a few biologically and physically meaningful parameters. Specifically, we use evolutionary conservation, physico-chemical properties and local/global geometry. These descriptors recently proved useful to identify and characterize protein-protein interfaces [32, 33]. JET$^2_{DNA}$ predicts surface patches following the *support-core-rim* model for experimental protein-protein interfaces, where interacting residues are classified based on their structural role in the interaction [34]. This three-layer model comes from a reexamination of the previously proposed *core-rim* one, which proved useful in dissecting protein-protein [35, 36] and

protein-DNA interfaces [37] and in understanding their properties. To the best of our knowledge, this is the first time that the three-layer *support-core-rim* model is used for analyzing and predicting protein-DNA interfaces. $\text{JET}^2_{\text{DNA}}$ implements three different scoring strategies, aimed at detecting different types of protein-DNA interfaces and different subregions of a given interface.

We assessed $\text{JET}^2_{\text{DNA}}$ performance on two new representative benchmarks of high-resolution structures, comprised of 187 protein-DNA complexes (HR-PDNA187) and 82 associated bound-unbound pairs (HOLO-APO82), and on an independent test set of 82 nucleic acid binding proteins (TEST-NABP82), taken from [18]. We show that $\text{JET}^2_{\text{DNA}}$ predictions are very accurate and robust to protein conformational and stoichiometry changes associated to DNA binding. Moreover, the predictive performances are equivalent for both DNA- and RNA-binding sites, showing that these sites share similar properties. In addition, rigorous comparison with three established machine learning based methods, namely DISPLAR [7], multiVORFFIP [15] and DRNApred [18] demonstrates the better predictive power of $\text{JET}^2_{\text{DNA}}$ on all datasets. Specifically, $\text{JET}^2_{\text{DNA}}$ is able to significantly detect more interacting residues while retaining similar precision and to unravel 'hidden' alternative binding sites. We show how one can learn about the origins and specificities of different types of protein-DNA interfaces through direct interpretation of $\text{JET}^2_{\text{DNA}}$ predictions. With this work, we pave the way to the discovery of yet unknown binding sites, opening up new perspectives for drug design.

We provide the structures and the experimentally known DNA-binding sites for our two newly created datasets, HR-PDNA187 and HOLO-APO82 at: http://www.lcqb.upmc.fr/PDNAbenchmarks/. These datasets represent the largest body of non-redundant known high-resolution crystallographic protein-DNA complex structures. They were manually curated to ensure quality and biological relevance. They can be used as benchmark sets for evaluating DNA-binding site predictors and DNA-protein docking methods. The code of $\text{JET}^2_{\text{DNA}}$ is available at: http://www.lcqb.upmc.fr/JET2DNA.

## Materials and methods

### Datasets

**HR-PDNA187.** The complete list of 1257 protein-double strand DNA complexes determined by X-ray crystallography with a resolution better than 2.5Å was downloaded from the Nucleic Acid Database [38] (February 2016 release). This list was filtered using PISCES sequence culling server [39] to define a set of 222 protein-DNA complexes non-redundant at 25% sequence identity, with an R-factor lower than 0.3 and with at least one protein chain longer than 40 amino acids. The complexes' 3D structures were downloaded from the PDB [4].

Subsequently, the dataset was manually curated to ensure its good quality. We removed entries where: (1) the asymmetric unit did not contain at least one biological unit or (2) the DNA molecule was single-stranded or contained less than 5 base pairs or (3) only C$\alpha$ atoms were present. Moreover, only chains with more than one contact with the DNA were retained. When the biological unit contained multiple copies of the protein-DNA complex, only one copy was kept. The structure 4hc9 was excluded because it displays different DNA-binding sites in the asymmetric and biological units. The complex 4aik was substituted with the 100% homolog 4aij, where the DNA-binding site is twice bigger. Finally, we removed the only antibody present in the dataset (3vw3), since this class of proteins have very peculiar characteristics and should be treated separately. In total, we retained 187 complexes (Table A in S1 File).

HR-PDNA187 covers all major groups of DNA-protein interactions according to Luscombe *et al.* classification [1]: helix-turn-helix (HTH), zinc-coordinating, zipper type, other $\alpha$-

helical, $\beta$-sheet, $\beta$-hairpin/ribbon, other. It spans a wide range of functional classes: it comprises 100 enzymes, 78 regulatory proteins, 7 structural proteins, 1 protein with other function and 1 unclassified protein (Table A in S1 File).

**HOLO-APO82.** We collected all available X-ray structures of the APO forms of the proteins from HR-PDNA187. We used the blastp program from the BLAST+ package [40] from NCBI with a threshold of 95% for seq. id., $10^{-3}$ for the E-value, a percentage of coverage $\geq$ 70% and a percentage of gaps $\leq$ 10% with respect to the query protein chain. Among the structures matching these criteria, only the ones having the same UniProt code [41] or belonging to the same organism as the query sequence were retained. If several structures passed all filters, only the closest one to the query or the one with the highest resolution was chosen. We found unbound forms for 81 complexes. For the complex 2isz, the unbound protein was found as a monomer and also as a dimer in two different PDB structures. Both structures were retained as they were both reported as present in equilibrium [42]. The resulting list comprises a total of 82 HOLO(bound)-APO(unbound) pairs. Within each pair, the APO form may be in the same oligomeric state as the HOLO form or may have fewer chains (Table A in S1 File).

**TEST-NABP82.** We used an additional recent dataset [18] of 82 proteins bound to nucleic acids, with a resolution better than 2.5Å. Among these, 49 proteins were solved bound to DNA (TEST-DBP49) and 33 to RNA (TEST-RBP33). Among the 49 DNA-binding proteins, 25 share more than 30% sequence identity with proteins from HR-PDNA187. To fairly assess $JET^2_{DNA}$ performance, they were removed from the dataset and the remaining 24 were used as non-redundant test set of DNA-binding proteins (TEST-DBP24). The list of the 24 complexes present in TEST-DBP24, annotated with the reasons they were excluded from HR-PDNA187 despite their good ($<$2.5Å) resolution, is reported in Table B in S1 File.

## Definition of interface residues

For each bound form from HR-PDNA187, we calculated the residues relative accessible surface areas in presence ($rasa_{DNA}$) and absence ($rasa_{free}$) of DNA, using NACCESS 2.1.1 [43] with a probe size of 1.4Å. Interface residues were defined as those being more buried in the presence of DNA than in the free form ($\Delta rasa > 0$). They were classified in three structural components [34]: *support* residues are buried both in presence ($rasa_{DNA}<$25%) and absence of DNA ($rasa_{free}<$25%); *core* residues are exposed in absence of DNA ($rasa_{free}\geq$25%) and become buried upon binding ($rasa_{DNA}<$25%); *rim* residues are exposed in presence ($rasa_{DNA}\geq$25%) and in absence ($rasa_{free}\geq$25%) of DNA (Fig 1a).

The interfaces for TEST-NABP82 were directly taken from [18]. They were defined from the PDB entries comprised in the dataset and also PDB structures of identical or similar complexes (seq. id.$\geq$80% and TM-score$\geq$0.5) [44, 45]. This allows to account for interface variability coming from molecular flexibility. Nevertheless, we should stress that these interfaces were defined using a very stringent distance criterion of 3.5Å, and hence they are substantially smaller than those we defined from HR-PDNA187. We used them to fairly assess $JET^2_{DNA}$ performance on an independent dataset, not created by us, and to fairly compare $JET^2_{DNA}$ with DRNApred [18], which was evaluated on them.

## Residue descriptors

**Evolutionary conservation.** Conservation levels are computed using the Joint Evolutionary Trees (JET) method [46]. This measure is inspired but different from the evolutionary trace introduced in [47, 48]. Briefly, the algorithm performs a PSI-BLAST search [49] to retrieve a set of sequences homologous to the query. These sequences are then sampled by a Gibbs-like approach and from each sample a phylogenetic tree is constructed [32, 46]. From

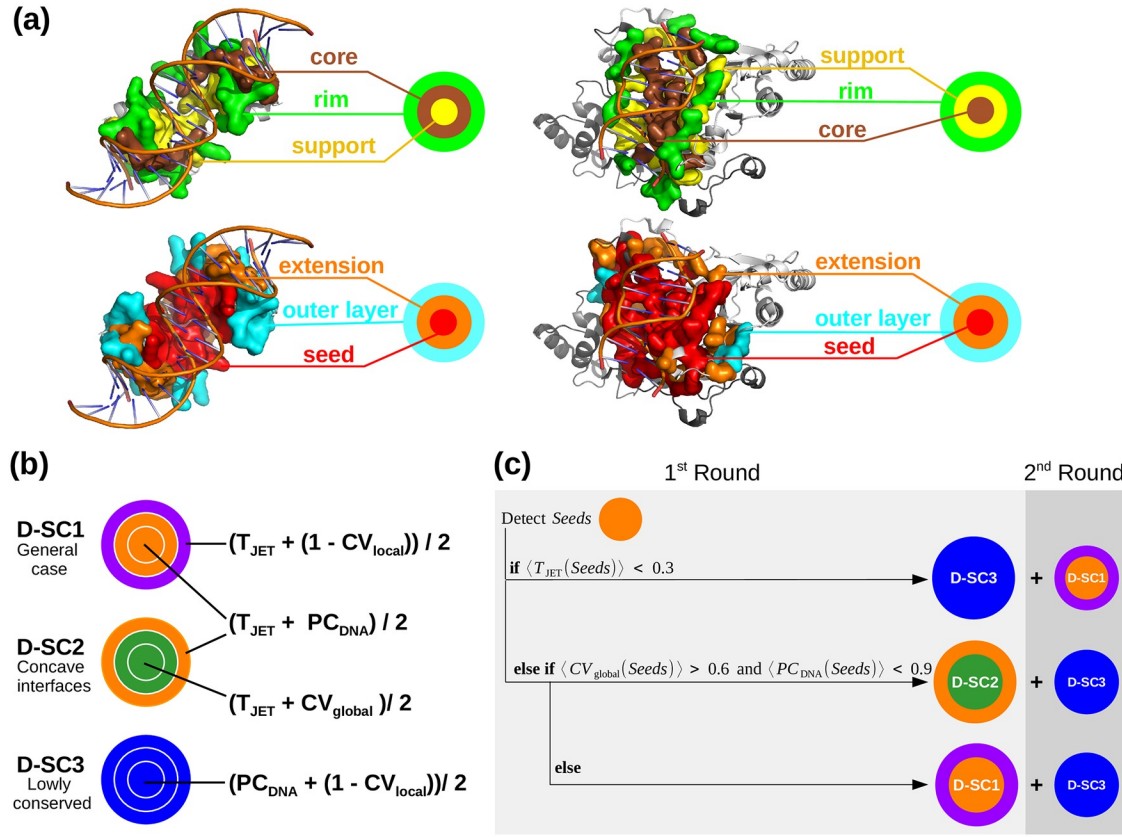

**Fig 1. Experimental interface definition, JET$^2_{DNA}$ scoring schemes and complete automated clustering procedure.** (a) Top, sections of two experimental interfaces (on the left, PDB code: 1JE8; on the right, PDB code: 1D02). Bottom, the corresponding JET$^2_{DNA}$ prediction using D-SC2. The experimental and predicted interface residues are displayed in opaque surfaces: *support*, *core* and *rim* are in yellow, brown and green, respectively; cluster *seed*, *extension* and *outer layer* are in red, orange and cyan, respectively; (b) Schematic representation of the three scoring schemes provided in JET$^2_{DNA}$. T$_{JET}$: conservation level, PC$_{DNA}$: protein-DNA interface propensities, CV$_{local}$ and CV$_{global}$: local and global circular variance computed with a radius of 12 Å and 100 Å, respectively. Different colors correspond to different combinations of properties used to predict interface residues in the three steps of the clustering procedure. (c) Schematic representation of the complete automated JET$^2_{DNA}$ clustering procedure.

each tree, a *tree trace* is computed for each position in the query sequence: it corresponds to the tree level where the amino acid at this position appeared and remained conserved thereafter [32, 46]. *Tree traces* are averaged over all trees to get more statistically significant values, denoted as T$_{JET}$, which vary in the interval [0, 1].

**Physico-chemical properties.** Propensities specific to every amino acid to be located at a protein-DNA interface (PC$_{DNA}$) were taken from [50] (see Fig. A in S2 File, in red). The original values, ranging from 0 to 2.534, were scaled between 0 and 1.

**Circular variance.** The circular variance (CV) is a geometrical measure of the vectorial distribution of a set of neighboring points localized around the center of a three dimensional sphere of radius $r_c$ [51, 52]. The CV value of an atom $i$ is computed as:

$$CV(i) = 1 - \frac{1}{n_i} \left| \sum_{j \neq i, r_i \leq r_c} \frac{\vec{r_{ij}}}{\|\vec{r_{ij}}\|} \right| \tag{1}$$

where $n_i$ is the number of all individual atoms distant by less than $r_c$ Å from atom $i$. The CV value of a residue is then computed as the average of the atomic CVs, over all its atoms.

Given a residue, its CV value reflects the protein density around it. A low CV value (close to 0) indicates that a residue is located in a protruding region of the protein surface, while a high value (close to 1) indicates it is buried within the protein. By consequence, the complement measure (1- CV) shows low values for buried residues and high values for protruding ones. As described in the following, since the algorithm clusters together highly scored residues, the $CV_{global}$ measure is then used to detect globally buried residues, while $(1 - CV_{local})$ is used to detect locally protruding ones.

Varying the radius cutoff $r_c$ allows describing the local ($CV_{local}$, $r_c$ = 12Å) and the global ($CV_{global}$, $r_c$ = 100Å) geometry of the surface. The values of $r_c$ for $CV_{local}$ and $CV_{global}$ are the same ones we already used in [32] to predict protein-protein interfaces. In [32], we showed that thresholds in the range 10-14Å for $CV_{local}$ would not change the results. On the other hand, a $CV_{global}$ value of 100Å ensures the inclusion of essentially all protein atoms in the calculation. Setting the threshold to a value larger than the standard protein size does not affect the efficiency nor the outcome of the calculation.

## $JET^2_{DNA}$ workflow

The $JET^2_{DNA}$ method requires as input a protein query sequence for which three-dimensional (3D) structural coordinates are available in the PDB. It computes $T_{JET}$, $PC_{DNA}$, $CV_{global}$ and $CV_{local}$ values and it combines them to assign a score to each residue. Different combinations are implemented in three different scoring schemes (D-SC in Fig 1b) designed to detect different types of protein-DNA interfaces (see Results). The computed scores are used to rank, select and cluster protein residues. The clustering algorithm is adapted from the protein-protein interface prediction method JET/$JET^2$ [32, 46]. First, highly scored residues proximal in 3D space are clustered together to form *seeds*. Then, the *seeds* are extended by progressively adding highly scored neighboring residues. At this stage, if two predicted patches are in contact, they will be merged. Finally, the prediction is completed with an *outer layer* of residues. The *seed*, *extension* and *outer layer* (Fig 1a, in red, orange and cyan, respectively) approximate the *support*, *core* and *rim* (Fig 1a, in yellow, brown and green, respectively) detected in experimental interfaces. For a more detailed description of the three-steps clustering procedure see Table C in S1 File. $JET^2_{DNA}$ implements an automated procedure to choose the most appropriate scoring scheme for the studied system (Fig 1c; see also *Automated clustering procedure*). Alternatively, the user can manually choose a scoring scheme. The user has the possibility to complement the predictions with another round of the clustering procedure using a complementary scoring scheme (Fig 1c; see also *Complete clustering procedure*). To get more robust predictions, several iterations of $JET^2_{DNA}$ can be run (i$JET^2_{DNA}$; see also *Iterative mode*). A schematic representation of the $JET^2_{DNA}$ pipeline is reported in Fig. B in S2 File. In the following, we explain in details the new criteria and new procedures implemented in $JET^2_{DNA}$, compared to $JET^2$ [32].

**Modification of the expected size of the interface.** At each step of the clustering procedure, $JET^2_{DNA}$ uses two thresholds, namely $score_{res}^{layer}$ and $score_{clus}^{layer}$, that respectively determine which residues should be considered as candidates for the predicted patches and when the process of growing each patch should be stopped. The set up of these threshold is based on the expected relative size of a protein-DNA interface, $f_{intfrac}^{DNA}(x)$. $f_{intfrac}^{DNA}(x)$ was determined by plotting the percentage of interface residues versus the total number of surface residues for HR-PDNA187 (Fig. C in S2 File, circles). The function that best approximates our data is $f_{intfrac}^{DNA}(x) = (2.66/\sqrt{x}) + 0.03$ (Fig. C in S2 File, solid line), where $x$ is the number of protein surface residues. Compared to protein-protein interfaces [32], DNA-binding sites cover a larger portion of the protein surface (Fig. C in S2 File, compare dotted and solid lines).

**Dynamic set up of the residue and cluster thresholds.** Contrary to JET² algorithm, where $score_{res}^{layer}$ and $score_{clus}^{layer}$ thresholds were fixed at the beginning of the clustering procedure and could not be changed, in JET²$_{DNA}$ we decided to initially fix them more stringently and relax them in a second stage, if the prediction is smaller than two thirds (<70%) of the expected size (Fig. B in S2 File, in red). This dynamic set up limits false positives in cases where surface regions outside but close to the interface display a detectable signal. It should be noted that the thresholds in JET²$_{DNA}$, even relaxed, remain stricter than the ones in JET². Details about threshold values used in JET²$_{DNA}$ are given in Table D in S1 File.

**Avoiding small-ligand binding pockets.** Like protein-protein interactions, protein-DNA interactions are often mediated or regulated by small ligands. As a result, a significant number of protein-protein and protein-DNA interfaces are close to or overlapping small ligand-binding pockets. These pockets are generally very conserved (*e.g.* active sites of enzymes) [32] and may thus confound the prediction. In JET², we resolved the issue by designing a specific scoring scheme exploiting the fact that small ligand-binding pockets are more deeply buried than protein-protein interfaces (see SC2 in Fig. 2 from [32]). The specific detection of protein-DNA interfaces is more difficult, as these interfaces are more concave than protein-protein interfaces (compare (1-CV) boxplots in Fig 2e–2g). To tackle the problem, we implemented a procedure in JET²$_{DNA}$ that redefines *seeds* when they are too buried. Specifically, if a *seed* comprises a significant proportion (>20% for D-SC1, >30% for D-SC2) of highly buried residues, then the clustering procedure restarts avoiding such residues (Fig. B in S2 File, in blue). The procedure does not apply to D-SC3 as this scoring scheme specifically selects protruding/exposed residues. Residues were considered as highly buried if their $CV_{local}$ was higher than $CV_{local}^{high} = 0.9$. They represent less than 3% of the protein surface (Fig. D in S2 File).

**Filtering out putative false positive clusters.** In JET², small patches were filtered out based on the comparison between their size and the size distribution of randomly generated patches [46] after the *seed* or *extension* steps, depending on the scoring scheme used. In the case of protein-DNA interfaces, we observed that this procedure removed too many true positives. Hence, we modified it in JET²$_{DNA}$. Namely, we first detect all possible *seeds* and extend them. Then, if the patches represents more than two thirds (>70%) of the expected size of the interface, we iteratively filter them starting from the smallest one (Fig. B in S2 File, in green). To eliminate aberrant predictions, clusters composed of 1 or 2 residues are still systematically filtered out.

## Automated clustering procedure

The implemented algorithm is described in Fig 1c and Table E in S1 File. By default, JET²$_{DNA}$ first detects *seeds* using D-SC1 ($T_{JET}$ + $PC_{DNA}$). If these display a very low conservation signal (average $T_{JET}$<0.3) then the strategy is to exploit the other two descriptors and look for locally protruding residues that satisfy expected physico-chemical properties (D-SC3). Otherwise, the algorithm analyses the physico-chemical and the geometrical properties of the detected *seeds*. If the *seeds* are in a very concave region (average $CV_{global}$>0.6) and do not display highly favorable physico-chemical properties (average $PC_{DNA}$<0.9), then the algorithm switches from D-SC1 to D-SC2, where $CV_{global}$ is employed to accurately detect an "enveloping" (concave) interface. Otherwise, physico-chemical properties are considered the driving force for an accurate prediction of the interface.

Selecting residues with a $CV_{global}$>0.6 allows to well define globally concave protein regions, giving a good compromise between too small and too large ones (see Fig. E(a) and Fig. E(b) in S2 File, on top).

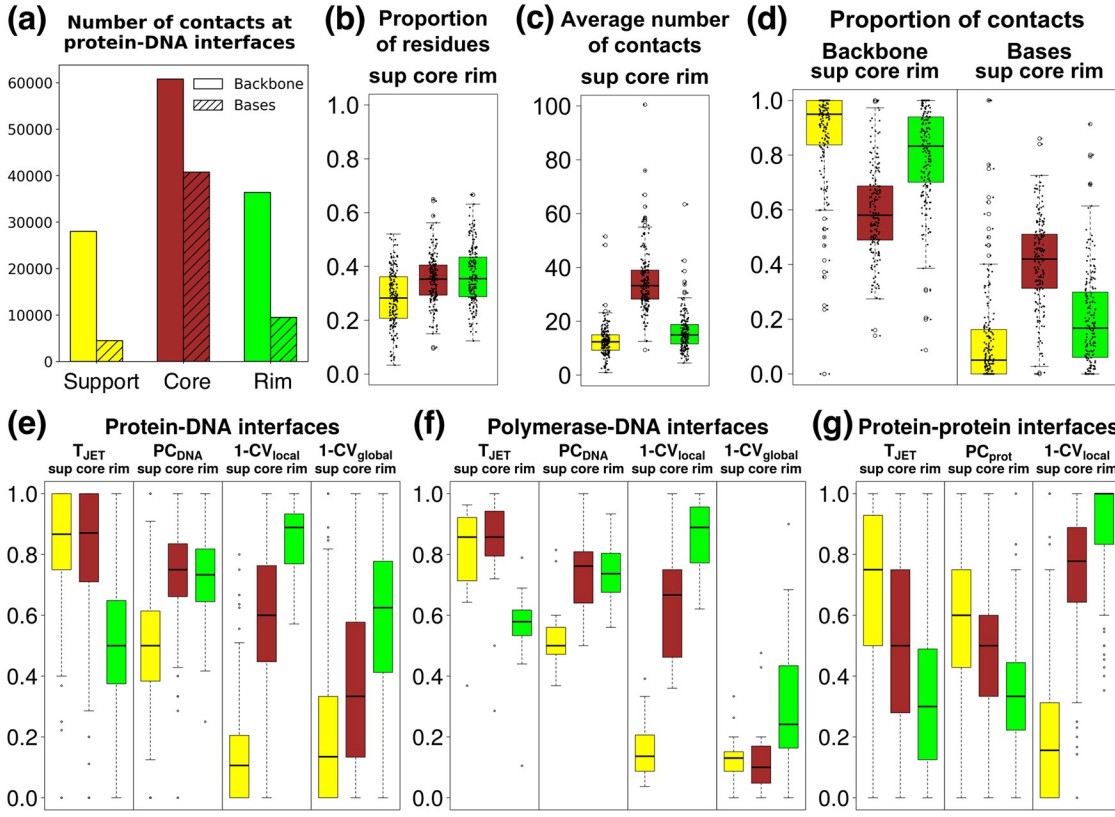

**Fig 2. Signals detected in experimental interfaces.** The calculations in (a)-(f) were performed on the HR-PDNA187 dataset. The plot in (g) was taken from Fig. 1a in [32]. The *support*, *core* and *rim* are in yellow, brown and green, respectively. Atomic contacts are defined by an atomic distance < 5Å. (a) Total number of atomic contacts between protein interface residues, divided in *support*, *core* and *rim*, and DNA backbone atoms (plain color) and DNA base atoms (diagonal hatching). (b) Distributions of the proportion of residues comprised in *support*, *core* and *rim*. (c) Distributions of the average number of atomic contacts per interface residue. Interface residues are divided in *support*, *core* and *rim*. One outlier point, reaching a value of 206, was removed in the boxplot of the distribution of *core* residues. (d) Distributions of the proportion of atomic contacts established by *support*, *core* and *rim* residues with DNA backbone atoms and DNA base atoms. (e-g) Distributions of the proportion of interface residues, divided in *support*, *core* and *rim*, having values above the median value computed over the entire protein. $T_{JET}$: conservation level, $PC_{DNA}$: protein-DNA interface propensities, $PC_{prot}$: protein-protein interface propensities, $CV_{local}$ and $CV_{global}$: local and global circular variances computed with a radius of 12Å and 100Å, respectively. Distributions are computed on: (e) all protein-DNA interfaces from HR-PDNA187, (f) all polymerases interfaces from HR-PDNA187, (g) all the 176 protein-protein interfaces in PPDBv4 (see Fig. 1a in [32]).

## Complete clustering procedure

A complete procedure is available for both manual and automated clustering procedure and is described in Fig 1c and Table E in S1 File (see also Fig. B in S2 File, in yellow). It consists in complementing the main clusters predicted in the first round by secondary clusters detected by another scoring scheme. D-SC3 will be used in the second round if D-SC1 or D-SC2 were chosen as main scoring schemes. D-SC1 will be the complementary one when D-SC3 is used in the first round.

## Iterative mode

Multiple $JET^2_{DNA}$ runs may lead to slightly different predictions, due to the Gibbs sampling of the sequences. To get more robust predictions, $JET^2_{DNA}$ can be run in an iterative mode of the program, which we call $iJET^2_{DNA}$. In this way, we can compute the number of times a given

residue is detected in a cluster divided by the total number of runs. The result will be a number comprised between 0 and 1 and it reflects the probability of the given residue to be at an interface.

## Evaluation of performances

We used six standard measures of performance:

$$Sens = \frac{TP}{TP + FN}; \quad PPV = \frac{TP}{TP + FP}; \quad Spe = \frac{TN}{TN + FP}; \quad Acc = \frac{TP + TN}{TP + FN + TN + FP};$$

$$F1 = \frac{2 \cdot Sens \cdot PPV}{Sens + PPV}; \quad MCC = \frac{TP \cdot TN - FP \cdot FN}{\sqrt{(TP + FP)(TP + FN)(TN + FP)(TN + FN)}}$$

where TP (true positives) are the number of residues correctly predicted as interacting, TN (true negatives) are the number of residues correctly predicted as non-interacting, FP (false positives) are the number of non-interacting residues incorrectly predicted as interacting and FN (false negatives) are the number of interacting residues incorrectly predicted as non-interacting. Thus, in our case, *Sensitivity* (*Sens*) measures the proportion of true interface residues that were correctly predicted as interacting with respect to the total number of interface residues; the *Positive Predictive Value* (*PPV*) measures the proportion of true interface residues that were correctly predicted as interacting with respect to the total number of predicted residues; the *Specificity* (*Spe*) measures the proportion of true non-interface residues that were correctly predicted as non-interacting with respect to the total number of non-interface residues; *Accuracy* (*Acc*) measures the proportion of correctly predicted residues (interacting and non-interacting) with respect to the total number of residues; *F1* score is a weighted average, specifically the harmonic mean, between *Sensitivity* and *PPV*, measuring the balance between these two; *Matthew's Correlation Coefficient* (*MCC*) is a correlation coefficient between observed and predicted residues. The *MCC* is the only measure varying in the range [-1, 1], where 0 represents a prediction no better than random. All the other measures return values in the range [0, 1]. We multiplied by 100 all the statistical values reported in tables and figures, representing them in percentages varying in the range [-100, 100] for the *MCC*, and [0, 100] for the other measures.

To assess the statistical significance of the differences in performance between each pair of methods, we relied on the paired *t*-test when the distributions were normal and on the Wilcoxon signed-rank test [53] otherwise. To verify if the data were normally distributed we used the Anderson-Darling test [54]. The difference was considered significant when the p-value was lower than 0.05.

## Choice of the other methods

To compare $JET^2_{DNA}$ performance, we considered a large set of popular DNA-binding site predictors [5–18, 20–24, 55, 56]. Among those, we discarded BindN [8], BindN+ [9], BindN-RF [12], MetaDBsite [55], PreDs [21], DBindR [13], PreDNA [10], DBD-Hunter [23], RBscore [20, 56] and PDNAsite [16] as their web servers are no longer available or do not work. We finally retained three methods with a relatively short runtime, namely DISPLAR [7], multiVORFFIP [15] and DRNApred [18]. DISPLAR was reported to show better performance than DP-Bind [11] and is more recent than DBS-Pred [6] and DBS-PSSM [5]. multiVORFFIP was chosen because it showed good performance on protein-protein interfaces prediction [32]. DRNApred is a very recent predictor that uses only sequence information and which aims at specifically detecting DNA-binding residues, discriminating them from RNA-binding residues. We ran the three tools with the default parameters. Since multiVORFFIP does not

provide binary prediction results (binding versus non-binding residues), predicted residues were defined by those with a normalized score (probability) $> 0.5$.

## Results

### *Support-Core-Rim* vs *Core-Support-Rim* model for protein-DNA interactions

To identify characteristic features of protein-DNA interfaces, we analysed the non-redundant set of protein structures bound to DNA available in the PDB. This dataset, HR-PDNA187, comprises 187 high-resolution crystallographic structures, covers all major types of DNA-interactions and spans a wide range of protein functions (see Materials and methods). Interface residues were classified based on the *support-core-rim* model (Fig 1a, on top) proposed for protein-protein interfaces [34]. The *support* (in yellow) comprises residues buried both in presence and absence of DNA, the *core* (in brown) comprises residues exposed in absence of DNA and becoming buried upon binding, and the *rim* (in green) comprises residues exposed both in presence and absence of DNA (see Materials and methods). In protein-protein interfaces, the three components are spatially organized in concentric layers, with the *support* at the center, the *core* in an intermediate position and the *rim* on the external border (see Fig. 2 in [32]). In protein-DNA interfaces, in addition to this spatial organization (Fig 1a, on top left), we also observe an organization where the *support* and the *core* switch positions (Fig 1a, on top right). This variety reflects the different ways a protein may bind to DNA.

However, *support*, *core* and *rim* residues seem to play the same role during the DNA binding in both structural organizations described above. Overall, *core* residues establish the majority of the atomic contacts (defined by an atomic distance$<5$Å) (Fig 2a and 2c) with both DNA backbone and DNA base atoms (Fig 2a), despite a comparable number of *support*, *core* and *rim* residues in HR-PDNA187 (Fig 2b) (*support*: 2501, *core*: 2958, *rim*: 2891). More than 75% of protein-DNA base contacts involve *core* residues (Fig 2a and 2d), while *support* and *rim* residues prevalently establish contacts with DNA backbone atoms (Fig 2a and 2d).

### Characteristics of protein-DNA interfaces

To characterize experimental protein-DNA interfaces, we relied on evolutionary conservation ($T_{JET}$), DNA-binding propensity ($PC_{DNA}$) and local and global burial degree ($CV_{local}$ and $CV_{global}$) (Materials and methods for precise definitions of the descriptors). Specifically, for each complex we computed the percentage of residues located in the *support*, *core* and *rim*, displaying higher values than the median computed over the entire protein (Fig 2e and 2f). The resulting distributions were compared with those obtained for the 176 protein-protein complexes of the Protein-Protein Docking Benchmark version 4 (PPDBv4) [57] (Fig 2g), already reported in Fig. 1a in [32].

DNA binding sites, especially the *support* and the *core*, are significantly more conserved than the rest of the entire protein (Fig 2e, $T_{JET}$). This conservation signal is stronger than in protein-protein interfaces (compare Fig 2e and 2g). Residues with high DNA-binding propensities tend to be located in the *core* and *rim* (Fig 2e, $PC_{DNA}$). By contrast, residues displaying physico-chemical properties favourable to protein-protein binding are mainly found in the *support*, and to a lesser extent in the *core*, of protein-protein interfaces (Fig 2g, $PC_{prot}$). These different geometrical distributions reflect differences between the two PC scales (Fig. A in S2 File). Protein-DNA interfaces are enriched in positively charged and polar residues, which tend to prefer regions exposed to the solvent. By contrast, protein-protein interfaces are enriched in hydrophobic residues, which prefer to be located towards the interior of the

interface. Regarding the surface geometry, the *core* and *rim* of protein-DNA interfaces are enriched in locally protruding residues (Fig 2e, (1-CV$_{local}$)), as observed for protein-protein interfaces (Fig 2g). When looking at the status of the interface with respect to the global shape of the protein surface (1-CV$_{global}$), we found that the enzymes, and in particular the polymerases and some nucleases, display a very specific profile (compare Fig 2e and 2f). All three interface components, especially the *support* and the *core*, are located in concave protein regions (Fig 2f), indicating that these proteins bind to DNA by "enveloping" it.

Overall, this analysis showed that protein-DNA interfaces encode signals that can be described by a few residue-based features and that the way these signals are distributed can be described using the *support-core-rim* model. It also confirmed that protein-DNA interfaces are more conserved than protein-protein interfaces, as reported in [37, 58, 59]. They display specific physico-chemical and geometrical characteristics that vary depending on the type of protein-DNA interaction considered. Hence, the correct detection of these interfaces requires the development of adapted scoring strategies.

### Three strategies to detect protein-DNA interfaces

Following the *support-core-rim* model (Fig 1a, on top), we use a predictive model comprising three components (Fig 1a, on bottom), namely the *seed* (in red), the *extension* (in orange) and the *outer layer* (in cyan). JET$^2_{DNA}$ implements different algorithms and scores to detect each one of these components (see Materials and methods). To be able to predict a wide range of protein-DNA interfaces, we devised three scoring schemes (see D-SC1-3, Fig 1b). Each D-SC combines T$_{JET}$, PC$_{DNA}$, CV$_{local}$ and CV$_{global}$ in a different way. As the *support* and the *core* may exchange their positions, the same combination is used for *seed* detection and *extension* in all D-SC (Fig 1b, same color for the two first layers). Specifically,

- **D-SC1** first detects highly conserved residues with good physico-chemical properties (high values of (T$_{JET}$ + PC$_{DNA}$)) and completes the prediction with conserved locally protruding residues (high values of (T$_{JET}$ + (1 − CV$_{local}$))). It aims at detecting generic DNA-binding sites.

- **D-SC2** clusters together conserved residues located in highly concave regions (high values of (T$_{JET}$ + CV$_{global}$)); then, the prediction is completed by adding an *outer layer* of conserved residues displaying good physico-chemical properties (high values of (T$_{JET}$ + PC$_{DNA}$)). It is designed to specifically detect interfaces characteristic of the "enveloping" binding mode displayed by polymerases.

- **D-SC3** leaves out conservation, focusing on locally protruding residues displaying good physico-chemical properties (high values of (PC$_{DNA}$ + (1 − CV$_{local}$))). It is intended to deal with cases where no evolutionary information is available or the whole protein displays a homogeneous conservation signal. Moreover, it is useful to complete predictions of the previous two scoring schemes, when some residues are less or no conserved than the rest of the binding site, sometimes because they are DNA sequence specific for a subfamily of the DNA-binding protein (see Fig 3, bottom panel, and *Discovery of alternative DNA-binding sites*).

Examples of predictions are shown on Fig 3. D-SC1 is particularly suited for double-headed interfaces stacking in the DNA grooves and some single-headed interfaces. These binding modes are often adopted by transcription factors, regulatory proteins and some glycosilases. The "enveloping" mode captured by D-SC2 is found in many polymerases and nucleases. D-SC3 deals with cases where physico-chemical and geometrical properties have a better discriminative power than conservation. By default, JET$^2_{DNA}$ will automatically determine the

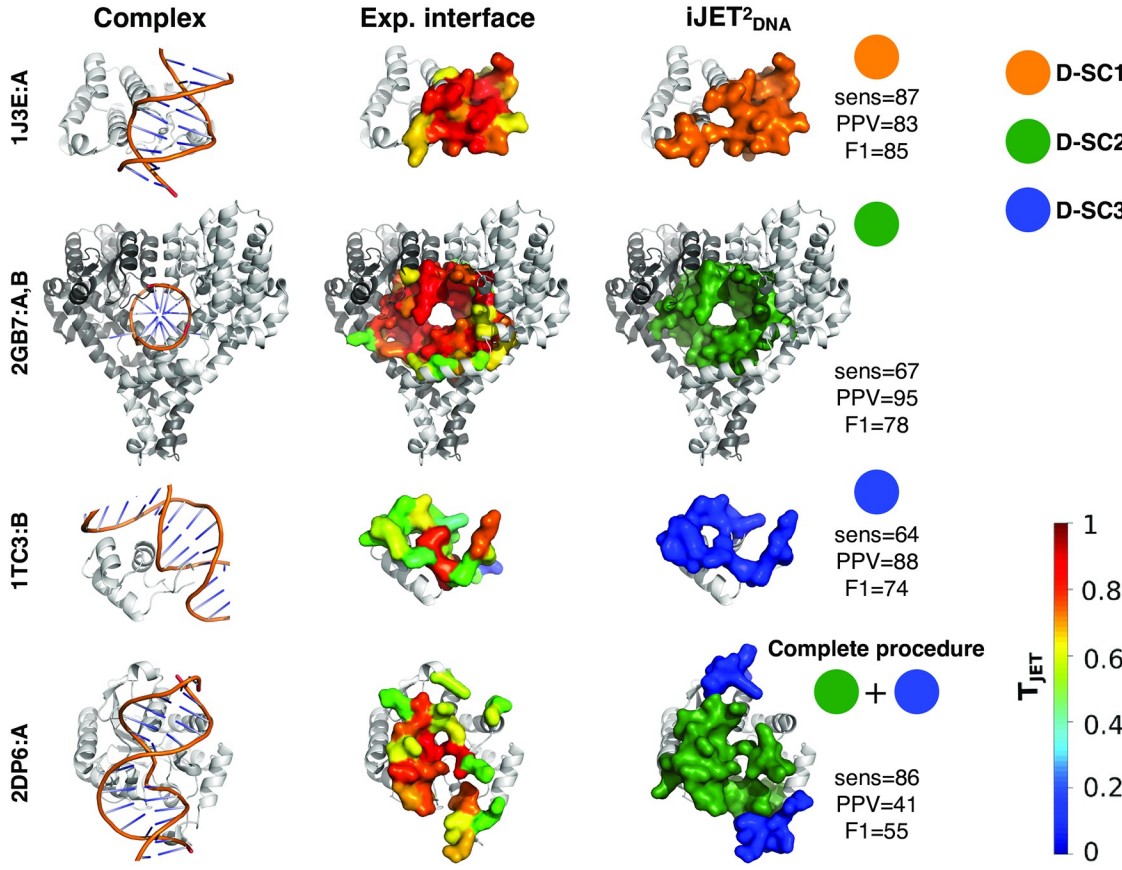

**Fig 3. Examples of DNA-binding sites predicted by iJET$^2_{DNA}$.** In the first column, the experimental complexes formed between the proteins of interest (greyscaled colored chains) and the DNA (orange) are represented as cartoons. In second and third columns, the experimental and predicted DNA-binding sites are displayed as opaque surfaces, respectively. The experimental interface residues are colored according to conservation levels (T$_{JET}$ values) computed by iJET$^2_{DNA}$. iJET$^2_{DNA}$ predictions, obtained from a consensus of 2 runs out of 10, are colored according to the scoring scheme: D-SC1 in orange, D-SC2 in dark green and D-SC3 in blue. Statistical performance values are given in percentages.

scoring scheme the most suited to the protein being analyzed (Fig 1c). In Fig. E in S2 File, we report five examples of predictions by JET$^2_{DNA}$ automated clustering procedure. In the first three cases, the protein concave regions coincide with the respective DNA-binding sites and the JET$^2_{DNA}$ automated clustering procedure is able to correctly choose, for the first and third cases, the D-SC2 scoring scheme designed to accurately predict these types of protein-DNA interfaces (Fig. E(a) in S2 File, on bottom). Concerning the second case of Fig. E(a) in S2 File, the JET$^2_{DNA}$ automated clustering procedure chose D-SC1, likely due to the physico-chemical properties of this region. However, the predicted region by D-SC1 mostly overlap with the D-SC2 one. In the other two cases, the concave regions are distant from the respective DNA-binding sites and the JET$^2_{DNA}$ automated clustering procedure is able to avoid them and predict the correct DNA-binding sites by choosing D-SC1 (Fig. E(b) in S2 File, on bottom), likely because the conservation and physico-chemical properties of the concave regions do not match with the ones expected for DNA-binding sites. Alternatively, the user can apply the scoring scheme of his/her choice.

Some DNA-binding sites may be comprised of several regions exhibiting different signals. To correctly detect those, JET$^2_{DNA}$ implements a *complete* procedure where patches predicted in

a first round are complemented by patches predicted in a second round and using a complementary scoring scheme (Fig 1c). An example of prediction is given in Fig 3 (bottom panel). It may also happen that the DNA-binding site is overlapping or in close proximity to a small ligand binding pocket. $JET^2_{DNA}$ implements a procedure to specifically avoid these pockets (see Materials and methods). The impact of such procedure can be appreciated on Fig. F in S2 File.

## Overall assessment of $JET^2_{DNA}$ performance

We report results obtained using the iterative mode of $JET^2_{DNA}$ ($iJET^2_{DNA}$, see Materials and methods), with a consensus of 2 and 8 iterations out of 10. For each protein, the best predicted patch or combination of patches among the three scoring schemes was retained for the evaluation. $iJET^2_{DNA}$ reaches an average F1-score of 61% on HR-PDNA187 and of 58-59% on a subset of 82 proteins (HOLO-APO82) for which unbound forms are available (Table F in S1 File and Fig 4, on top). The performance is similar on bound and unbound forms, indicating that $JET^2_{DNA}$ is robust to conformational changes associated with DNA-binding. For the vast majority of proteins (82%), the sensitivity attained on the unbound form is at least as high as 90% of the sensitivity on the bound form, and in some cases it is even higher than 100%. Hence, $JET^2_{DNA}$ is able to detect interacting residues even when they are 'hidden' by conformational changes. Moreover, the extent of the conformational change is not correlated with the difference in performance (Fig. G-I in S2 File). The predictions are also robust to stoichiometry changes (Table G in S1 File). Varying the consensus threshold enables shifting the balance between sensitivity (Sens) and precision (or predictive positive value, PPV), such that the lower threshold (2/10) yields more extended predictions while the higher threshold (8/10) yields more precise ones. The *seeds* on their own cover about a third of the experimental

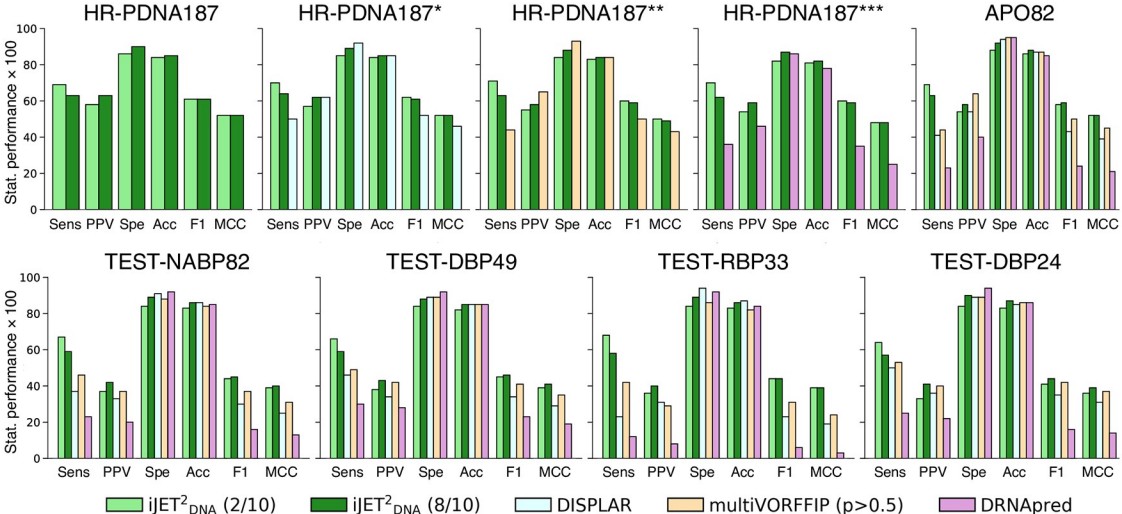

**Fig 4. Comparison of $iJET^2_{DNA}$, DISPLAR, multiVORFFIP and DRNApred performance.** For $iJET^2_{DNA}$, consensus predictions were obtained from 2 (in light green) and 8 (in dark green) runs out of 10. Statistical performance values are given in percentages. **Top panel**. HR-PDNA187 and APO82. To fairly assess DISPLAR, multiVORFFIP and DRNAPred performances, the proteins used for training these methods were removed from HR-PDNA187. We used a sequence identity cutoff of 95% and ended up with 106 (*), 87 (**) and 42 (***) proteins, respectively. **Bottom panel**. TEST-NABP82 and its subsets. TEST-DBP49 and TEST-DBP24 contain only DNA-binding sites, while TEST-RBP33 contains only RNA-binding sites. TEST-DBP24 and TEST-RBP33 contain only proteins sharing less than 30% sequence identity with proteins from HR-PDNA187 (see Materials and methods). DRNApred and multiVORFFIP were evaluated on their RNA-specific algorithms for the RNA-binding proteins, and on their DNA-specific ones for the DNA-binding proteins. $iJET^2_{DNA}$ and DISPLAR, which do not have a specific algorithm to predict RNA-binding sites, were evaluated on the same algorithm for both DNA- and RNA-binding proteins.

interfaces and contain very few positives (Table H in S1 File). On an independent test set comprising both DNA- and RNA-binding sites (TEST-NABP82), iJET$^2_{DNA}$ achieves an average F1-score of 45 (Table I in S1 File and Fig 4, on bottom). The sensitivity is roughly the same as for HR-PDNA187 but the precision (PPV) is lower. This can be explained by the fact that the experimental reference interfaces are defined using a much more stringent distance cutoff (see Materials and methods). iJET$^2_{DNA}$ detects equally well DNA- and RNA-binding sites (Fig 4, on bottom, compare TEST-DBP49/24 and TEST-RBP33).

## Discovery of alternative DNA-binding sites

The complete and/or automated mode(s) of JET$^2_{DNA}$ produce patches that do not match the experimental interfaces present in our datasets (Table J in S1 File). Due to the relatively small number of protein-DNA complexes available in the PDB, it is difficult to systematically assess the pertinence of these additional predictions. Nevertheless, we could identify several cases where the protein binds to DNA via two completely distinct binding sites. We report four such cases in the following (Fig 5 and see Table K in S1 File for F1 values).

1. **RNA polymerase from bacteriophage T7**. This protein first binds the DNA promoter via its recognition site [26, 60] (*site 1*, Fig 5a, left), then undergoes a large conformational change and starts the transcription at its catalytic active site [27] (*site 2*, Fig 5b, left), where a short strand of RNA is paired to the DNA strand. We considered the ensemble of nucleic acid binding residues as experimental interface. In both structures, D-SC1 (and D-SC2 with comparable values) correctly detected the catalytic active site (Fig 5a and 5b, right, in orange) while D-SC3 predicted the recognition site and some DNA-binding residues flanking the active site (Fig 5a and 5b, right, in blue). The fact that the two sites are detected by two different D-SC indicates that they are characterized by different properties, which correlate with their respective functions. Indeed, the active site is generic while the recognition site is specific of this protein family [26]. The very high conservation signal of the former masks the weaker signal of the latter (compare colors in Fig 5a and 5b, left), which is still detectable based on the other properties.

2. **N-terminal domain of the adeno-associated virus replication protein**. This protein contains three distinct DNA-binding sites [61]: a stem loop sequence specific binding site (*site 1*, Fig 5c, left), a tetranucleotide repeat recognition site (*site 2*, Fig 5d, left), and the Tyr153 active site (no PDB structure). D-SC1 and D-SC3 lead to accurate predictions of *site 1* and *site 2* (Fig 5c and 5d), which both display rather low conservation signal and are mostly protruding/exposed.

3. **Modification-dependent restriction endonuclease**. In the original structure from HR-PDNA187 (Fig 5e, left), the DNA is bound only to the winged-helix domain binding site [62] (*site 1*), while the catalytic domain (*site 2*) is partially disordered. In the alternative structure (Fig 5f, left), both DNA-binding sites are occupied. Although the relative domain orientations differ drastically between the two structures, JET$^2_{DNA}$ accurately identified the two binding sites by combining D-SC1 with D-SC3 (Fig 5e and 5f, right, in orange and in blue). The fact that a combination of D-SC is required reflects the heterogeneity of the conservation signal within each site (Fig 5e and 5f, left, colored by conservation level). D-SC3 enables rescuing lowly conserved subregions that D-SC1 is not able to detect.

4. **Cyclic GMP-AMP synthase**. DNA binding to this protein is associated to a catalytic reaction producing a cyclic dinucleotide from ATP and GTP. The protein was solved bound to DNA both in monomeric (Fig 5g, left [63]) and dimeric (Fig 5h, left [28]) forms. Since the

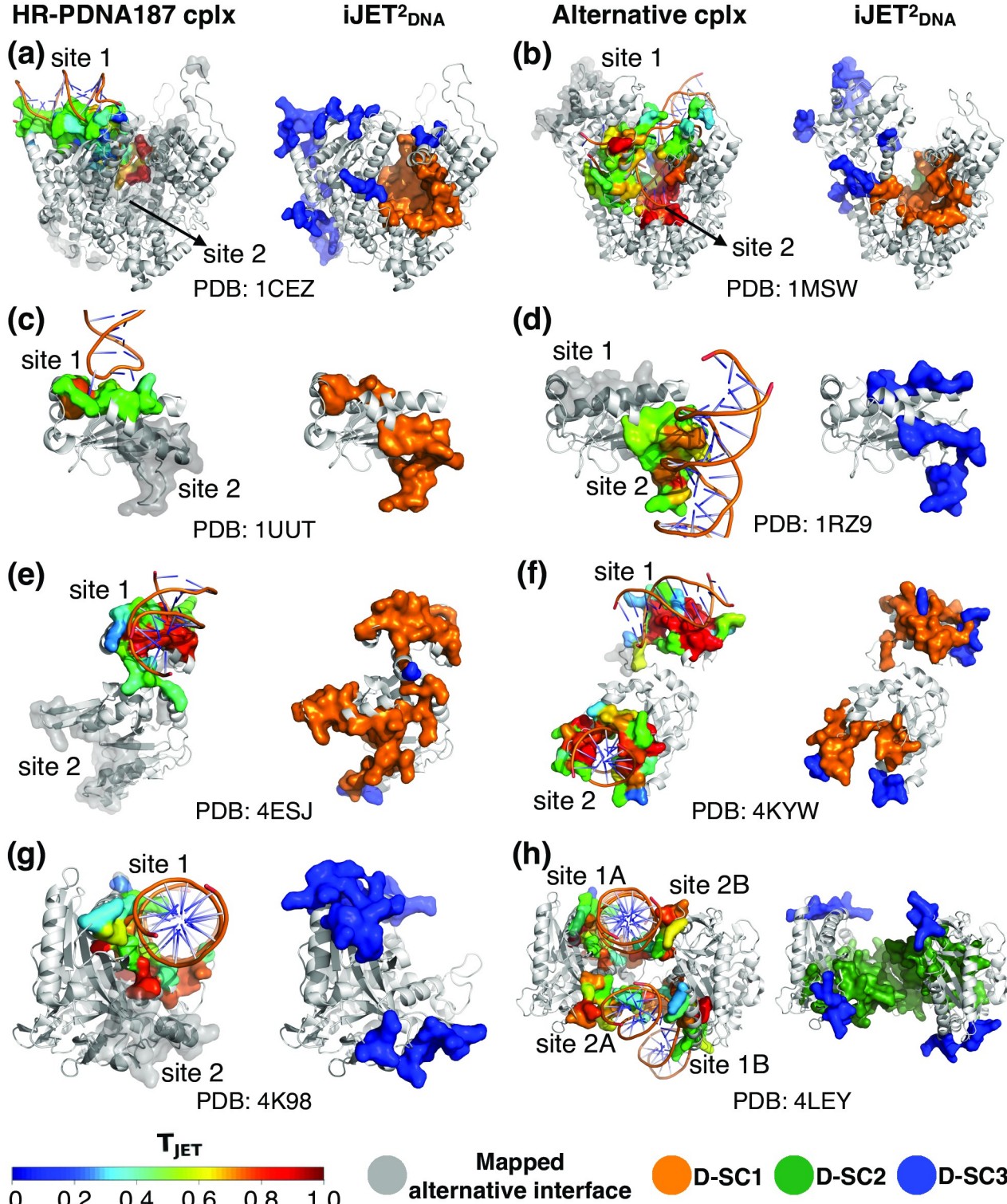

**Fig 5. Prediction of multiple DNA-binding sites by iJET$^2_{DNA}$.** For each protein (each row), the experimental complex present in HR-PDNA187 and another experimental complex displaying a distinct DNA-binding site are shown on the first and third columns, respectively. For each structure, the 'main' DNA-binding site is colored according to T$_{JET}$ values and the site coming from the other structure is mapped and displayed in transparent grey. The iJET$^2_{DNA}$ predictions computed on each experimental structure are displayed on the second and fourth columns, respectively, and colored according to the scoring scheme. (a-b) RNA polymerase from bacteriophage T7 (PDB: 1CEZ and 1MSW); (c-d) N-terminal domain of the adeno-associated virus replication protein (1UUT and 1RZ9); (e-f) R.DpnI modification-dependent restriction endonuclease (4ESJ and 4KYW); (g-h) Cyclic

GMP-AMP synthase (cGAS) (4K98 and 4LEY). iJET$^2_{DNA}$ predicted patches were obtained from a consensus of 2, for (a-b) and (e-f), or 5, for (c-d) and (g-h), runs out of 10. See Fig. J-L in S2 File and Table K in S1 File for comparison with other tools.

small ligand-binding active site is much more conserved than the DNA-binding sites, this case is particularly difficult. In the monomeric form, only D-SC3 is able to identify the two DNA-binding sites (Fig 5g, right, in blue). Upon dimerization, the interface geometry changes such that the two sites form a large concave region (Fig 5h, left), detected by D-SC2. Complementing the prediction with D-SC3 further improves the prediction (Fig 5h, right, in dark green and in blue).

These examples illustrate the different usages of JET$^2_{DNA}$ multiple scoring schemes and how one can learn about each DNA-binding site properties from JET$^2_{DNA}$ predictions. Specifically, different D-SC may target different binding sites (recognition or catalytic), or different regions of the same binding site (conserved *core* or protruding *rim*), or even yield overlapping predictions. In all cases, JET$^2_{DNA}$ detection proved to be robust to extensive conformational changes and stoichiometry changes.

## Comparison with other tools

We compared JET$^2_{DNA}$ with three other state-of-the-art machine learning based predictors, namely DISPLAR [7], multiVORFFIP [15] and DRNApred [18] (see Materials and methods). iJET$^2_{DNA}$ achieves ∼20% higher sensitivity and similar accuracy compared to DISPLAR and multiVORFFIP on all the datasets (Table F and Table I in S1 File and Fig 4). The predictions are slightly less precise (by ∼5-10%) than those of multiVORFFIP on HR-PDNA187** and APO82, but equally or more precise than DISPLAR and multiVORFFIP on the other datasets (Table F and Table I in S1 File and Fig 4). This results in ∼5-10% higher F1 values and most of the times ∼5-10% higher MCC values compared to both tools (Table F and Table I in S1 File and Fig 4). Compared to DRNApred, iJET$^2_{DNA}$'s detection is at least twice more sensitive and at least ∼10% more precise on all the datasets (Table F and Table I in S1 File and Fig 4). We should stress that DRNApred was designed to specifically discriminate DNA- from RNA-binding residues, which is not the purpose of JET$^2_{DNA}$. In particular, JET$^2_{DNA}$ performs much better than the three other tools on the RNA-binding proteins of TEST-RBP33 (Table I in S1 File and Fig 4). Moreover, when looking at the four above mentioned cases of multiple DNA-binding sites, JET$^2_{DNA}$ proved more powerful to detect alternative binding sites compared to the three other tools (Fig. J-L in S2 File and Table K in S1 File).

## Comparison with other benchmarks

We compared HR-PDNA187 and HOLO-APO82 with the most popular benchmarks, namely PDNA62 [6, 64], DBP374 [13], MetaDBsite316 [55], Displar264 [7], PDDB1.2 [65], DBP206 [66], PDNA224 [10] and DNABINDPROT54 [24]. Contrary to our datasets, most of them comprise only single chains, even if the functional biological unit of the protein in complex with DNA is annotated as a multimer. Moreover, when applying to them the PISCES criteria used to construct HR-PDNA187, the number of resulting complexes was systematically smaller than 187 (Table L in S1 File). Finally, very few of them provide the APO forms of the proteins.

## Robustness of the results to parameter changes

JET$^2_{DNA}$ parameters were set empirically based on our previous experience with JET and JET$^2$, our intuition and our analysis of the properties encoded in the experimental interfaces from

HR-PDNA187. However, we should stress that we did not use machine learning to infer them. Hence, HR-PDNA187 is not a training set *per se*. Moreover, $JET^2_{DNA}$ parameters are few, physically explained and biologically intuitive. To validate our choices, we conducted a thorough analysis of the impact of varying parameters on the predictions (Fig. M-W in S2 File). In total, we ran about 220 000 additional $JET^2_{DNA}$ calculations to assess the influence of seven parameters. These include the thresholds (% of residues and $CV_{local}$) used in the detection of small ligand pockets (Fig. Q-W in S2 File), the residue and cluster thresholds used to detect and grow patches (Fig. M-P in S2 File) and the confidence threshold used to filter out small patches (Fig. N-P in S2 File). This analysis showed that our choice of parameters is relevant, as the performances obtained with our default values are close to the best one can expect within the $JET^2_{DNA}$ framework. Moreover, we found that our default values are consistent with the intervals/regions they fall in and that there are no abrupt changes in performance within these regions. Hence, our predictions are stable to small parameter changes.

## Discussion

We have collected and carefully curated 187 high resolution protein-DNA complexes representative of all known types of protein-DNA interactions. This new dataset, supplemented by the 82 available protein unbound conformations, could serve as a reference benchmark for the community. Based on the analysis of the evolutionary and structural properties of the DNA-binding sites from this dataset, we have proposed an original method to predict them. Importantly, we do not estimate interacting probabilities for individual residues, but we predict ensembles of residues proximal in 3D space, in other words "patches". Predictions based on patches are justified by the fact that residues being conserved or displaying specific physico-chemical properties at protein-DNA interfaces tend to cluster together [58, 67, 68]. Moreover, we propose a "discretized model" of interfaces describing the role of each predicted residue in the interface. This model is inspired by the *support-core-rim* model defined for protein-protein interfaces. We have shown that it is also useful for protein-DNA interfaces, where the *support* and the *core* may switch their positions. We have defined three archetypal protein-DNA interfaces, namely conserved generic, conserved enveloping and not conserved, and have devised three scoring schemes to detect them.

We have thoroughly assessed our method's performance on bound and unbound protein forms and on an independent dataset. $JET^2_{DNA}$ compares favorably with machine learning based state-of-the-art methods. Specifically, the predictions are more sensitive and remarkably robust to extensive conformational changes and to stochiometry changes. Moreover, we have highlighted several cases where the same complex was solved in different conditions or the same protein was able to bind DNA via different locations. $JET^2_{DNA}$ was able to detect the 'alternative' binding sites despite major conformational changes between the different structures bound to DNA. A single crystallographic structure may reveal only one site or may even comprise a truncated or misplaced DNA, resulting in a "partial" associated binding site. In this context, $JET^2_{DNA}$ can be used as a mean to get a more realistic description of known binding sites, when those are only partially covered experimentally, and to discover yet unknown binding sites.

Beyond predicting DNA-binding sites, $JET^2_{DNA}$ provides a unique way to understand the origins and properties of these sites and interpret those in light of their functions. Transcription factors typically display single- or double-headed binding modes [69], with one or two highly conserved binding sites, which are well detected by D-SC1. Enzymes usually have larger interfaces to accommodate an exposed recognition site, detected by D-SC3, and a highly conserved active site, detected by D-SC1, or highly segmented protein-DNA interfaces, where the

protein interacts with the DNA through multidomain units in addition to their active site [25, 69, 70]. Moreover, $JET^2_{DNA}$ is useful to unravel the heterogeneity of signals comprised within given binding sites and to partition them in subregions displaying coherent properties. The predictions can help designing or repurposing small molecules to target protein-DNA interfaces in an intelligent way, *e.g.* specifically targeting the non-conserved subregions to avoid side effects.

There is growing experimental evidence that protein surfaces are used in many different ways by many partners. For example, for protein-protein interactions, it is now clear that several proteins can use the same region at the surface of a partner, possibly in different conformations, that binding sites can be overlapping and that different binding site properties relate to different interaction "types" or "functions" [33]. For protein-DNA interactions, the available structural data is much smaller. Still, the examples we showed let us envision a much larger complexity in the usage of protein surfaces by DNA than expected. Our work is inscribed in an effort to decipher such complexity.

Finally, we found that $JET^2_{DNA}$ can be successfully applied to the prediction of RNA-binding sites. This is in line with the growing body of evidence showing that proteins that bind DNA are also likely to bind RNA. Indeed, although DNA-binding proteins used to be considered as functionally different from RNA-binding proteins and studied independently, this view has become outdated [29]. There are many examples of proteins binding both nucleic acids either via the same region at different times or simultaneously via distinct regions. Some crystallographic structures of complexes between proteins and hybrid D/RNA molecules are also available in the PDB. Hence, discriminating DNA- from RNA-binding residues is very challenging. The DRNApred method [18] represents a recent effort to address this issue. We found that it predicts very few residues, compared to $JET^2_{DNA}$. A future development for $JET^2_{DNA}$ could be to include some RNA-specific features toward a better prediction of RNA-binding sites. It could be advantageous to include disorder information, an electrostatic potential in addition or replacing the residue propensities and/or restricting the calculation of the evolutionary conservation to the subfamily associated to the query protein. This could help in identifying family- or even protein-specific interfaces.

## Supporting information

**S1 File. Supplementary tables.**
(PDF)

**S2 File. Supplementary figures.**
(PDF)

## Acknowledgments

We thank Chloé Dequeker for sharing her scripts, N. Fernandez-Fuentes for indicating good practice for using the multiVORFFIP method and DISPLAR's authors for helping us in running the tool.

## Author Contributions

**Conceptualization:** Richard Lavery, Elodie Laine, Alessandra Carbone.

**Data curation:** Flavia Corsi.

**Funding acquisition:** Alessandra Carbone.

**Methodology:** Elodie Laine, Alessandra Carbone.

**Software:** Flavia Corsi, Elodie Laine.

**Supervision:** Elodie Laine, Alessandra Carbone.

**Validation:** Flavia Corsi.

**Writing – original draft:** Flavia Corsi, Elodie Laine, Alessandra Carbone.

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
