## [Decision Letter · Decision Letter 0]

14 Nov 2019

Dear Dr Carbone,

Thank you very much for submitting your manuscript, 'Multiple protein-DNA interfaces unravelled by evolutionary information, physico-chemical and geometrical properties', to PLOS Computational Biology. As with all papers submitted to the journal, yours was fully evaluated by the PLOS Computational Biology editorial team, and in this case, by independent peer reviewers. The reviewers appreciated the attention to an important topic but identified some aspects of the manuscript that should be improved.

We would therefore like to ask you to modify the manuscript according to the review recommendations before we can consider your manuscript for acceptance. Your revisions should address the specific points made by each reviewer and we encourage you to respond to particular issues Please note while forming your response, if your article is accepted, you may have the opportunity to make the peer review history publicly available. The record will include editor decision letters (with reviews) and your responses to reviewer comments. If eligible, we will contact you to opt in or out.raised.

- Supporting Information uploaded as separate files, titled 'Dataset', 'Figure', 'Table', 'Text', 'Protocol', 'Audio', or 'Video'.

We hope to receive your revised manuscript within the next 30 days. If you anticipate any delay in its return, we ask that you let us know the expected resubmission date by email at ploscompbiol@plos.org.

Sincerely,

Bert L. de Groot

Associate Editor

PLOS Computational Biology

Arne Elofsson

Deputy Editor

PLOS Computational Biology

[LINK]

Reviewer's Responses to Questions

**Comments to the Authors:**

Reviewer #1: Many different kinds of protein interact with nucleic acids to perform a variety of essential functions (eg enzymes and transcription factors); only a small fraction of the known 3D structures in the Protein Data Bank are of such complexes. This manuscript makes an important contribution to the field. It describes a computational method to identify and predict protein sites that can bind DNA or RNA molecules in a way that makes it possible to interpret the most prominent features underpinning the recognition. The performance of the methods is tested, and compared to other methods, using a carefully composed set of experimentally characterized protein-D/RNA complexes. Somewhat surprisingly the method, which was trained to detect DNA-binding sites, does quite well also regarding RNA-binding sites.

I have only a couple of minor items that the authors should look at:

The six performance measures are only presented as mathematical expressions (page 8) - to help the reader it would be useful to also explain what they measure, and how they relate to each other.

What is on the y-axis in Fig 4 (% of what?)?

Reference 32 is incomplete.

Reviewer #2: uploaded

**Have all data underlying the figures and results presented in the manuscript been provided?**

Reviewer #1: Yes

Reviewer #2: Yes

PLOS authors have the option to publish the peer review history of their article (what does this mean?). If published, this will include your full peer review and any attached files.

Reviewer #1: No

Reviewer #2: No

---

## [Editor Report · Decision Letter 1]

20 Dec 2019

Dear Dr Carbone,

We are pleased to inform you that your manuscript 'Multiple protein-DNA interfaces unravelled by evolutionary information, physico-chemical and geometrical properties' has been provisionally accepted for publication in PLOS Computational Biology.

In the meantime, please log into Editorial Manager at https://www.editorialmanager.com/pcompbiol/, click the "Update My Information" link at the top of the page, and update your user information to ensure an efficient production and billing process.

One of the goals of PLOS is to make science accessible to educators and the public. PLOS staff issue occasional press releases and make early versions of PLOS Computational Biology articles available to science writers and journalists. PLOS staff also collaborate with Communication and Public Information Offices and would be happy to work with the relevant people at your institution or funding agency. If your institution or funding agency is interested in promoting your findings, please ask them to coordinate their releases with PLOS (contact ploscompbiol@plos.org).

Thank you again for supporting Open Access publishing. We look forward to publishing your paper in PLOS Computational Biology.

Sincerely,

Bert L. de Groot

Associate Editor

PLOS Computational Biology

Arne Elofsson

Deputy Editor

PLOS Computational Biology

---

## [Editor Report · Acceptance letter]

24 Jan 2020

PCOMPBIOL-D-19-01575R1 

Multiple protein-DNA interfaces unravelled by evolutionary information, physico-chemical and geometrical properties

Dear Dr Carbone,

I am pleased to inform you that your manuscript has been formally accepted for publication in PLOS Computational Biology. Your manuscript is now with our production department and you will be notified of the publication date in due course.

With kind regards,

Matt Lyles
